# MicroRNAs as Predictive Biomarkers in Patients with Colorectal Cancer Receiving Chemotherapy or Chemoradiotherapy: A Narrative Literature Review

**DOI:** 10.3390/cancers15051358

**Published:** 2023-02-21

**Authors:** I-Ping Yang, Kwan-Ling Yip, Yu-Tang Chang, Yen-Cheng Chen, Ching-Wen Huang, Hsiang-Lin Tsai, Yung-Sung Yeh, Jaw-Yuan Wang

**Affiliations:** 1Department of Nursing, Shu-Zen College of Medicine and Management, Kaohsiung 82144, Taiwan; 2Division of Colorectal Surgery, Department of Surgery, Kaohsiung Medical University Hospital, Kaohsiung Medical University, Kaohsiung 80708, Taiwan; 3Department of Surgery, Faculty of Medicine, College of Medicine, Kaohsiung Medical University, Kaohsiung 80708, Taiwan; 4Division of Pediatric Surgery, Department of Surgery, Kaohsiung Medical University Hospital, Kaohsiung Medical University, Kaohsiung 80708, Taiwan; 5Graduate Institute of Clinical Medicine, College of Medicine, Kaohsiung Medical University, Kaohsiung 80708, Taiwan; 6Division of Trauma and Surgical Critical Care, Department of Surgery, Kaohsiung Medical University Hospital, Kaohsiung Medical University, Kaohsiung 80708, Taiwan; 7Department of Emergency Medicine, Faculty of Post-Baccalaureate Medicine, College of Medicine, Kaohsiung Medical University, Kaohsiung 80708, Taiwan; 8Graduate Institute of Injury Prevention and Control, College of Public Health, Taipei Medical University, Taipei 11031, Taiwan; 9Graduate Institute of Medicine, College of Medicine, Kaohsiung Medical University, Kaohsiung 80708, Taiwan; 10Center for Cancer Research, Kaohsiung Medical University, Kaohsiung 80708, Taiwan; 11Pingtung Hospital, Ministry of Health and Welfare, Pingtung 90054, Taiwan

**Keywords:** microRNAs, colorectal cancer, chemoresistance, radioresistance, predictive biomarkers

## Abstract

**Simple Summary:**

Nearly two decades would be required for a cancer lesion to develop from normal colon mucosa, but most colorectal cancer (CRC) patients are at an advanced stage at presentation. Chemotherapy, targeted therapy, and radiotherapy can improve the prognosis of patients with advanced CRC, but sometimes the therapy resistance occurs, and the 5-year survival rate for patients with locally advanced CRC and with metastatic CRC (mCRC) remain poor. MicroRNAs (miRs) can regulate cancer pathways by inhibiting their target mRNA translation and triggering their degradation. MiRs can serve as predictive biomarkers for the detection of CRC or mCRC or the resistance of chemotherapy or chemoradiotherapy, and miRNA-based therapeutics may finally reach the clinical stages.

**Abstract:**

Colorectal cancer (CRC) is one of the most common malignancies and is associated with high mortality rates worldwide. The underlying mechanism of tumorigenesis in CRC is complex, involving genetic, lifestyle-related, and environmental factors. Although radical resection with adjuvant FOLFOX (5-fluorouracil, leucovorin, and oxaliplatin) chemotherapy and neoadjuvant chemoradiotherapy have remained mainstays of treatment for patients with stage III CRC and locally advanced rectal cancer, respectively, the oncological outcomes of these treatments are often unsatisfactory. To improve patients’ chances of survival, researchers are actively searching for new biomarkers to facilitate the development of more effective treatment strategies for CRC and metastatic CRC (mCRC). MicroRNAs (miRs), small, single-stranded, noncoding RNAs, can post-transcriptionally regulate mRNA translation and trigger mRNA degradation. Recent studies have documented aberrant miR levels in patients with CRC or mCRC, and some miRs are reportedly associated with chemoresistance or radioresistance in CRC. Herein, we present a narrative review of the literature on the roles of oncogenic miRs (oncomiRs) and tumor suppressor miRs (anti-oncomiRs), some of which can be used to predict the responses of patients with CRC to chemotherapy or chemoradiotherapy. Moreover, miRs may serve as potential therapeutic targets because their functions can be manipulated using synthetic antagonists and miR mimics.

## 1. Introduction

Colorectal cancer (CRC) is a major public health problem among both sexes and has a worldwide mortality rate of 47.8% [1,2]. According to the annual report of the Health Promotion Administration in USA, each year, more than 1.1 million patients receive diagnoses of CRC worldwide and approximately 608,000 CRC-related mortalities occur, making CRC the third most common cause of death [1]. Approximately 20–30% of patients with stage I–III CRC who undergo a surgical resection eventually develop distant metastasis, which is associated with poor prognoses [3]. Researchers must continue investigating biomarkers that can be used to more accurately identify patients with CRC who are at a high risk of recurrence.

Adjuvant or neoadjuvant FOLFOX chemotherapy, which involves the use of 5-fluorouracil (5-FU), leucovorin (LV), and oxaliplatin, is widely used to reduce the risk of recurrence in patients with advanced-stage CRC [4,5,6]. Even after undergoing radical surgical resection or oxaliplatin-based adjuvant chemotherapy, some patients ultimately develop recurrence or metastasis, indicating that current treatments for CRC are insufficient [4,7,8,9]. Genomic and metabolomic analysis of right-sided and left-sided CRC are keystones in the study and treatment of subtypes of CRC [10]. In patients with metastatic CRC (mCRC), two signaling pathways—the epidermal growth factor receptor (EGFR) and vascular endothelial growth factor (VEGF) pathways—are involved in the proliferation and metastasis of CRC [11]. FOLFIRI (5-FU, LV, and irinotecan) plus anti-EGFR monoclonal antibodies provides a survival benefit to patients with mCRC with distant metastases [12,13]. For patients with right-sided mCRC, FOLFIRI or FOLFOXIRI (5-FU, LV, irinotecan, and oxaliplatin) plus bevacizumab (an anti-VEGF monoclonal antibody) are preferred first-line treatment options, irrespective of the patient’s RAS and B-Raf proto-oncogene, serine/threonine kinase (BRAF) mutational statuses [14]. For patients with locally advanced rectal cancer (LARC), neoadjuvant chemoradiotherapy (NACRT) is a standard treatment that can improve the outcomes of radical resection, prevent local recurrence, ensure sphincter preservation and tolerable toxicity levels, and maintain postsurgical quality of life [15,16,17,18]. However, patient responses to NACRT are highly variable [19,20,21], and resistance to chemoradiotherapy (CRT) is a major obstacle in the treatment of patients with LARC [22].

In patients with CRC, epigenetic modifications, including DNA methylation and histone modifications, can be used as clinical biomarkers for diagnosis, prognosis, and the prediction of patient responses to adjuvant or neoadjuvant therapy [23,24,25]. Liquid biopsies employ a wide range of technologies to acquire tumor information, including levels of carcinoembryonic antigen (CEA), circulating tumor cells (CTCs), circulating tumor DNA (ctDNA), circular RNAs (circRNAs), and microRNAs (miRs) in body fluids [26]. CEA is widely used as a surrogate biomarker in clinical practice for treatment response prediction and follow-ups in patients with CRC [27,28]. Moreover, the serum levels of CEA and expression levels of EGFR and FGD5-AS1 (FGD5 Antisense RNA 1, an oncogenic long non-coding RNA) have been observed to be significantly elevated in 5-FU-resistant CRC cells [29,30]. Using CTCs as individualized biomarkers can help healthcare providers develop effective treatment strategies for patients with CRC. The persistent presence of CTCs after adjuvant chemotherapy indicates chemoresistance and is often reflected in subsequent recurrence of CRC [31,32]. In addition, the progression of CRC can be modulated via circRNAs, which can sponge and downregulate target miRs [33]. In CRC, miRs have been determined through gene network analysis to act as both oncogenes and tumor suppressors of differentially expressed mRNAs and proteins [34,35,36,37].

MiRs, small noncoding RNAs consisting of approximately 20 nucleotides, can posttranscriptionally regulate the expression of several target genes by directly binding to the 3′ untranslated regions (3′-UTRs) of target mRNAs, thereby triggering mRNA degradation, suppressing mRNA translation, and subsequently regulating the protein expression [38]. The heterogeneity of CRC makes its accurate classification and treatment challenging [39]. The specific panel of miRNAs/mRNAs or miR clusters can contribute to the progression from adenomatous and carcinomatous lesions [40,41]. Four consensus molecular subtypes (CMSs) were identified via the classification of patients with CRC in biologically homogeneous CRC subtypes, and mesenchymal CMS4 presented with a worse prognosis [39]. Plasma miR profiling has been observed to be significantly associated with colorectal cancer CMS and validated for predicting both prognosis and treatment response [42]. For example, the miR-200 family is the most powerful determinant of CMS4-specific gene expression, which is associated with epithelial–mesenchymal transition [39]. The analysis of serum miRNAs is less invasive and still facilitates a real-time analysis of the disease course [42]. To date, researchers have identified at least 250 miRs that may serve as diagnostic biomarkers as well as prognostic indicators of CRC [43]. Therefore, miRs have the potential to serve as biomarkers for cancer detection, prognosis, CMS classification, and treatment response prediction [42,43,44,45,46,47]. MiR panels have been reported to facilitate early detection of relapse in patients with CRC [48,49,50,51].

MiRs are attractive as biomarker candidates because they can be identified through liquid biopsies, which makes them minimally invasive and more convenient for use in the early detection of relevant signals [52]. Several reviews have discussed miRs as biomarkers in CRC [53,54,55], and even textbooks mention the potential cancer-sensitizing agents for CRC chemotherapy [56]. Stiegelbauer et al. (2014) summarized the chemotherapeutic approaches for treating CRC and highlighted the role of miRNAs as predictive biomarkers for chemoresistance in patients with CRC [53]. In their meta-analysis, Masuda et al. concluded that miRs have strong statistical confidence as biomarkers for the early detection of CRC and prediction of prognosis and chemoresistance, but they also indicated that most miR reports were small-scale studies until 2017 [54]. Shirafkan et al. summarized the roles of miRs in CRC by emphasizing their importance in different signaling pathways, such as the EGFR, transforming growth factor beta (TGF-β), and tumor protein (TP53) pathways, and suggested miRs as predictive factors of chemotherapy [55]. Elucidating the mechanisms underlying the development of resistance to chemotherapy and radiotherapy is crucial to the development of more effective cancer treatment strategies. However, no reviews have focused on miRs as predictors of response to chemoradiotherapy in patients with CRC.

Numerous studies have explored the activation of regulatory pathways involving miRs in response to chemotherapy, radiotherapy, or antitumor agents, and miRs are attractive candidates for targeted therapy because their functions can be manipulated through the use of synthetic antagonists and miR mimics [43,44,45,46,47]. No miRNA-based drugs are currently on the market; however, many RNA-based therapies, including antisense oligonucleotides, aptamers, small interfering RNAs, miRs, and mRNA, are currently undergoing clinical trials or have already received regulatory approval (including as treatments for liver cancer, lymphoma, and melanoma) [57]. Identifying a reliable biomarker for predicting chemoresistance or radioresistance in patients with CRC may improve survival outcomes [58,59,60]. In this review, we would discuss how these miRs can serve as clinical biomarkers in the early detection of CRC or mCRC and as predictive biomarkers in CRC treatment.

## 2. Methods

We searched PubMed, the Cochrane Central Register of Controlled Trials, and the Cochrane Library for relevant studies. We did not apply any language or regional restrictions. We used the following MeSH terms in our search: (“Colorectal Neoplasm”[Title/Abstract]) AND (MicroRNA*[Title/Abstract]) AND (“Chemotherapy”[Title/Abstract] OR “Chemoradiotherapy”[Title/Abstract] OR “Neoadjuvant Radiotherapy [Title/Abstract] OR “Neoadjuvant Chemotherapy*”[Title/Abstract]). We thoroughly evaluated all the relevant studies and bibliographies to identify additional potentially eligible studies.

## 3. Results

### 3.1. MiRs Associated with the Detection of CRC and mCRC

MiRs may act as oncogenes or tumor suppressors and may serve as biomarkers for the early diagnosis of CRC to facilitate efficient treatment (Figure 1). The expression levels and functions of miRs in various body extracts (including serum, tumor tissues, feces, or urine) have been analyzed in several case–control studies (Table 1). Expression levels of miRs in feces or urine can be used for noninvasive screening for the detection of CRC. Researchers developed a urinary biomarker panel combining miR-129-1-3p and miR-566 that could accurately detect stage 0/I CRC [48]. The expression levels of miR-29a, miR-223, and miR-224 in fecal samples from patients with CRC were all significantly lower than those in fecal samples from healthy volunteers (all *p* < 0.001) [61]. A systematic review revealed that the expression levels of miR-20a in the feces, serum, or tumor tissues of patients with CRC were upregulated relative to those in the control samples, indicating that miR-20a may serve as an accurate biomarker for CRC detection [62]. MiR-106a and miR-125b are associated with the pathogenesis of CRC and may therefore also be used as significant prognostic markers of early-stage CRC [63]. The overexpression of miR-21 in CRC tumor tissue was significantly associated with advanced CRC, but interestingly, the lower expression levels of serum miR-21 were associated with a higher mortality rate [64,65].

MiRs are used not only in the diagnosis of primary CRC but also in the prediction of early relapse of CRC. In several case–control studies including patients with UICC stage II and III CRC, the miR-29c expression levels in the tumor tissues of the early relapse group were significantly lower than those in the tumor tissues of the non–early relapse group [66]. Tsai et al. reported that serum miR-148a expression in the early relapse group was significantly lower than that in the non–early relapse group [67,68]. In another study, lower miR-148a expression in CRC tissues was positively associated with an advanced TNM stage, poor tumor differentiation, lymph node metastasis, and distant metastasis [69].

Researchers have identified some miRs that can be used to predict the development of metastasis in CRC. Chen et al. reported that plasma miR-96/miR-99b can be used as a biomarker for the early detection of mCRC [72]. In another study, the circulating miR-762 levels of patients with CRC with distant metastasis were higher than those of patients with CRC without distant metastasis [74]. Pidikova and Herichova discovered that the miR-17/92a-1, miR-106a/363, miR-106b/93/25, and miR-183/96/182 clusters were strongly associated with metastasis and poor patient survival [70]. Hoye et al. analyzed various next-generation sequencing data sets of samples of primary CRC and mCRC (liver, lung, and peritoneal metastases) and tumor-adjacent tissues and identified five miRNAs—miR-210-3p, miR-191-5p, miR-141-3p, miR-1307-5p, and miR-155-5p—that were upregulated at multiple metastatic sites [73], which may serve as distinguishing biomarkers of mCRC.

Several miR panels have been established on the basis of case–control studies. For example, Wang S. et al. developed a serum panel of miR-409-3p, miR-7, and miR-93 that could accurately discriminate between the plasma samples of patients with CRC and healthy controls [75]. Another serum miR panel comprising miR-203a-3p, miR-145-5p, miR-375-3p, and miR-200c-3p could also discriminate between the plasma samples of patients with CRC and healthy controls with a sensitivity of 81.25% and a specificity of 73.33% [76]. Another panel combining six clinicopathologic factors with six miRs (miR-7, miR-93, miR-195, miR-141, miR-494, and let-7b) could be used to detect early relapsed CRC with a sensitivity of 89.4% and a specificity of 88.9% [49].

### 3.2. MiRs Associated with the Prediction of Responses to Chemotherapy in CRC

Resistance to chemotherapy is one of the most common reasons for treatment failure among patients with CRC, and many patients with advanced CRC are initially responsive to chemotherapy but ultimately develop chemoresistance [77,78]. OncomiRs, which can increase chemoresistance, may reduce the levels of apoptosis proteins and silent apoptosis information and neutralize or reverse antiapoptotic signals [54].

Table 2 lists oncomiRs that can enhance chemoresistance through specific regulatory pathways. The transcription factor p53 is the most thoroughly characterized tumor suppressor gene, and p53 variants appear in approximately 50% of patients with CRC [79]. Loss or mutation of the p53 gene and its related pathways, such as those involving *TP53INP1* and *TP53INP2* or Bcl-2 and caspase proteins, is positively associated with therapeutic resistance in various cancers [80,81,82,83]. Several oncomiRs, including let-7f-5p [84,85], and miR-96 [86], are overexpressed in patients with chemoresistant CRC (relative to patients with chemosensitive CRC) (Table 2). MiR-34a may serve as a predictor of 5-FU chemosensitivity in CRC, and a combination of miR-34a and 5-FU is effective as a treatment for CRC [87]. Chemoresistance may be related to the AMP-activated protein kinase-mammalian target of the rapamycin (AMPK–mTOR) pathway, and several miRs, including miR-27a, miR-103, and miR-107, are overexpressed in patients with chemoresistant CRC (relative to patients with chemosensitive CRC) [88,89,90,91]. Overexpression of miR-744 may mediate oxaliplatin chemoresistance in CRC by suppressing *BIN1* expression [92]. Chemoresistance was correlated with cancer cell stemness in patients with CRC [69,89,90], and patients expressing the CD44 variant appear to present with more aggressive phenotypes of CRC. Moreover, miR-1246 was overexpressed in CD44v6+ cells and was associated with poor overall survival and disease-free survival in patients with CRC [93].

Table 3 presents anti-oncomiRs that can modulate patients’ sensitivity to chemotherapy. Overexpression of miR-141-3p negatively regulates epithelial–mesenchymal transition (EMT) and can restore chemosensitivity; in a previous study, the 5-year overall survival of the high miR-141-3p expression group was superior to that of the low miR-141-3p expression group [94,95,96]. Overexpression of miR-377-3p (miR-154 family) or miR-193b-5p also enhanced chemosensitivity to 5-FU in CRC cells by negatively regulating EMT or the forkhead box M1-ATP-binding cassette subfamily C member 5 (FOXM1-ABCC5/10) signaling pathway and decreasing cell stemness [97,98,99,100]. Xiao et al. demonstrated that miR-1915-3p can improve the chemotherapeutic efficacy of oxaliplatin in CRC cells by suppressing EMT-promoting oncogenes, 6-phosphofructo-2-kinase/fructose-2,6-biphosphatase 3 (*PFKFB3*), and ubiquitin specific peptidase 2 (*USP2*) [101].

The mechanism of 5-FU-induced cytotoxicity in CRC cells involves the inhibition of thymidylate synthase (TS), a 5-FU target enzyme. In one analysis of clinical CRC samples, the miR-375 expression levels in the 5-FU-resistant group were much lower than those in the 5-FU-sensitive group [107]. MiR-375 can enhance 5-FU cytotoxicity in CRC cells by suppressing TS and the Sp1 transcription factor (*SP1*) [107,108]. High miR-218 and miR-330 expression levels had a positive prognostic value in 5-FU-based treatments for CRC, and miR-218 inhibits TS [105,106]. In one study, lower miR-148a expression was associated with advanced CRC and distant metastasis, and overexpression of miR-148a suppressed the expression of stem cell markers and increased chemosensitivity by regulating the β-catenin signaling pathway [69]. In a study by Huang et al., upregulated miR-148a enhanced chemoradiosensitivity and promoted apoptosis of CRC cells by targeting the MET proto-oncogene, receptor tyrosine kinase (c-Met), in vitro and in vivo [113]. High miR-148a expression levels were associated with more favorable tumor responses to neoadjuvant CRT and survival outcomes [113]. By conducting a Kaplan–Meier survival analysis of a sample of 62 patients, researchers determined that miR-27b-3p expression levels are positively correlated with disease-free survival [102]. The MYC proto-oncogene, bHLH transcription factor (c-Myc), can downregulate miR-27b-3p expression, inducing oxaliplatin resistance in CRC cells by inhibiting autophagy [102].

Diabetes mellitus and hyperglycemia have been demonstrated to affect chemoresistance in and the prognosis of CRC [114]. In one study, miR-488 mimic decreased glucose uptake and increased oxaliplatin/5-FU-sensistivity in CRC cells by targeting the oncogene, 6-phosphofructo-2-kinase/fructose-2,6-biphosphatase 3 (*PFKFB3*) [109]; the miR-488 expression levels of patients with metastatic/recurrent CRC were significantly lower than those of patients with CRC without metastasis/recurrence; and low miR-488 expression levels were associated with low 3-year survival rates, poor differentiation, and advanced-stage disease [109]. Another study demonstrated that tumor-secreted miR-208b promotes Treg expansion by targeting programmed cell death factor 4 (*PDCD4*) and may be related to chemoresistance to oxaliplatin in CRC [115]. In one study, the diagnostic accuracy (in terms of area under the curve [AUC]) of CEA and CA19-9 (for distinguishing between oxaliplatin-chemoresistant and oxaliplatin-chemosensitive patients) was 0.542 and 0.686, respectively, whereas a panel containing miR-100, miR-92a, miR-16, miR-30e, miR-144-5p, and let-7i achieved the highest diagnostic accuracy, with an AUC of 0.825 [116]. Comparing plasma miR levels with CEA and CA19-9 in the follow-up of patients with CRC, CEA, and CA19-9 showed a higher specificity for CRC but a lower sensitivity than miRs in predicting disease recurrence [117].

EGFR expression serves as a prognostic factor in patients with stage III CRC receiving metronomic maintenance therapy [29]. In one study, the inhibition of EGFR by the specific inhibitor erlotinib effectively enhanced the antitumor toxicity of 5-FU though miR-330 directly targeting thymidylate synthase [30]. For patients with mCRC with *RAS* variations, one therapeutic alternative is targeting and preventing angiogenesis by inhibiting VEGF. FOLFIRI plus bevacizumab can increase sensitivity to and enhance the antitumor effects of 5-FU (Table 3) [118]. Upregulation of miR-1207-5p can suppress bevacizumab resistance in bevacizumab-resistant CRC cells by modulating the expression of ATP-binding cassette subfamily C member 1 (*ABCC1*) [110]. Lower miR-1287-5p expression levels upregulate the expression of multifunctional *Y-box binding protein 1* (*YBX1*) and are associated with cetuximab resistance in CRC [111,112]. Circulating plasma levels of miR-20b, miR-29b, and miR-155 serve as predictors of bevacizumab efficacy in patients with mCRC (Table 1) [71]. In one study, the serum miR-148a expression levels of patients with mCRC who exhibited partial responses to treatment were higher than those of patients with mCRC with disease progression [103]. MiR-148a decreases angiogenesis and increases the apoptosis of CRC cells by downregulating hypoxia-inducible factor 1 subunit alpha (HIF-1α)/VEGF and the MCL1 apoptosis regulator, BCL2 family member (Mcl-1), and serum miR-148a levels have prognostic and predictive value in patients with mCRC receiving bevacizumab therapy [103].

### 3.3. MiRs Associated with the Prediction of Responses to Radiotherapy in CRC

Radiotherapy induces various DNA lesions and unpaired double-strand breaks; therefore, biomolecules that inhibit DNA repair pathways usually improve radiosensitivity [119,120]. Radiation-induced DSBs principally activate the intrinsic apoptotic pathway, and two principal methods are involved in DSB repair: homologous recombination (HR) and nonhomologous end joining (NHEJ) [121]. MiRs are some of the most important mediators of radiosensitivity in CRC cells and are involved in the HR and NHEJ repair pathways [122,123]. Huang et al. reviewed and discussed the biological functions of miRNAs in the regulation of radiosensitivity in CRC cells [123]. However, some investigations of radioresistance in CRC cells were in vitro studies in which CRC cell lines were transfected with miR mimics or knocked out to modulate the expression of various miRs, such as miR-101-3p [124], miR-185 [125], miR-195 [126], and miR-5197 [127]; these miRs were omitted from Table 4.

By analyzing the differentially expressed miRNA profiles of patients with CRC, researchers have identified several oncomiRs that enhance radioresistance as biomarkers of responses to radiotherapy (Table 4). p21 is a tumor suppressor gene that can protect cancer cells from DNA damage [128,129]. The overexpression of several miRs, including miR-106b and miR-222, can inhibit p21 expression, thereby inducing radioresistance through the phosphatase and tensin homolog/phosphatidylinositol-3 kinase/AKT serine/threonine kinase 1 (PTEN/PI3K/AKT) pathway, in vitro and in vivo [104,130]. MiR-93 and miR-106b are categorized into the same family, and upregulated miR-93 induces radioresistance by downregulating forkhead box A1 (FOXA1) and upregulating transforming growth factor beta 3 (TGFB3) [131,132,133]. Exosomes containing miR-93-5p derived from cancer-associated fibroblasts can prevent CRC cells from undergoing radiation-induced apoptosis [131]. MiR-96-5p also reportedly induces radioresistance in rectal cancer cells by inhibiting glypican 3 (GPC3) and abnormally triggering the canonical Wnt signaling pathway (Table 4) [134]. MiR-19b targets the F-box and WD repeat domain containing 7 (*FBXW7*), thereby promoting CRC stem cell stemness and inducing radioresistance, and in one study, patients with LARC with low miR-19b expression levels had markedly longer overall survival and event-free survival [135,136].

**Table 4 cancers-15-01358-t004:** OncomiRs that enhance radioresistance in CRC and mCRC.

Family	MiRs	Verified Targets in CRC or Other Cancers	Sample Source	Target for miRNA	Ref.
miR-17	miR-17, miR-18a/b, miR-106a/b, miR-20a/b, miR-93	MiR-106b induces cell radioresistance by targeting the PTEN/PI3K/AKT pathways and *p21* in CRC. MiR-93 acts as a specific exosomal cargo that increases radioresistance. Inhibition of miR-93 suppressed radioresistance.	Tumor tissue Tumor tissue	PTEN/PI3K/AKT pathways and *p21* *FOXA1* *BTG3*	[130] [131] [133]
miR-19	miR-19a, miR-19b-1, miR-19b-2	The low miR-19b expression levels of patients with LARC had markedly longer OS and DFS. MiR-19b induces radioresistance, and the patients with higher miR-19b expression levels had a shorter survival time.	Tumor tissue	*FBXW7*	[135] [136]
miR-96	miR-96	miR-96-5p induced radioresistance is upregulated in rectal cancer cells through the inhibition of *GPC3* and abnormal triggering of the canonical Wnt signaling pathway.	Tumor tissue	*GPC3*	[134]
miR-103	miR-103a/b, miR-107	MiR-107 induces chemoresistance. Hsa-mir-107 and WDFY3-AS2 may serve as prognostic biomarkers in RC.	Tumor tissue	Through the CAB39–AMPK–mTOR pathway	[89] [91]
miR-221	miR-221, miR-222	MiR-222 induces radiation resistance.	Serum	*PTEN*.	[104]

Likewise, high miR-125b expression levels in tissue and serum were associated with a poor treatment response in patients with LARC [137] (Table 5). By analyzing receiver operating characteristic curves, D’Angelo et al. demonstrated that circulating miR-125b levels exhibited greater discriminatory power for treatment responses than serum CEA levels did; therefore, miR-125b may serve as a new noninvasive predictive biomarker in the treatment of LARC [137]. Recent clinical findings have demonstrated that cancer stem cells and their inherent radioresistance are crucial to local control after radiotherapy [138]. Differentially expressed miRs, such as miR-148a and miR-214, were observed in radiated CRC cells. MiR-214 promotes radiosensitivity by inhibiting autophagy-related 12 (*ATG12*)-mediated autophagy in CRC [139,140]. Comparisons of the serum miR expression levels of patients with CRC before or after radiation therapy have revealed that radiation therapy can significantly reduce serum miR-296-5p expression. Moreover, miR-296-5p overexpression significantly reduced the survival fraction of CRC cells under ionizing radiation (IR) treatment by targeting the insulin-like growth factor 1 receptor (IGF1R) and musashi RNA binding protein 1 (MSI1) [141,142]. Therefore, miRs serve as potential regulators of radioresistance in CRC cells and, in turn, may be a promising therapeutic target in the treatment of CRC.

### 3.4. MiRs Associated with the Prediction of Responses to Chemoradiotherapy in CRC

The present study focused on oncomiRs and anti-oncomiRs modulating the sensitivity of chemoradiotherapy, and we noted that studies on chemotherapy- or radiotherapy-related miRs far outnumber those on chemoradiotherapy-related miRs (Table 6). Huang et al. demonstrated that alterations in miR-148a overexpression may enhance chemoradiosensitivity and promote apoptosis by directly targeting c-Met in vivo in both human CRC cells and mice [113]. Moreover, miR-34a attenuates chemoradioresistance in CRC [87,143]. *MRX34*, a miR-34a mimic, is the first synthetic miR to undergo clinical trials to restore the sensitivity of CRC cells to chemotherapeutic agents [143].

### 3.5. Current Undergoing Clinical Trials for miRs

After the discovery of the dysregulation of miRNA expression itself being associated with human disease progress and therapy response, the introduction of a disease suppressor miRNA mimic to restore its functionality is one therapeutic approach under careful consideration [144]. To date, dozens of miRNA molecules are in clinical trials [144], such as: miravirsen (miR-122) for the treatment of hepatitis C virus (HCV) infection, under phase II clinical trials in several countries, Remlarsen (miR-29) for the treatment of different type of fibrosis, under phase I clinical trials, and MRX34 (miR-34a) for the treatment of different types of cancers [144,145]. MRX34 is a synthetic, double-stranded miR-34a mimic that encapsulated in a liposomal nanoparticle for the treatment of different types of advanced solid tumors [145,146]. In the first-in-human Phase 1 study of MRX34-based cancer therapy, Hong et al. (2020) revealed that the need to anticipate toxic effects for this class of miR-based drug and the effective delivery of these RNA constructs also remains an unresolved challenge [145].

## 4. Conclusions

As highlighted in this review, miRs have considerable potential as predictive biomarkers in patients with CRC receiving chemotherapy or chemoradiotherapy; therefore, miRs warrant the further investigation in prospective clinical studies. The clinical roles of miRs in the regulation of chemosensitivity and radiosensitivity in CRCs must be further explored. By interacting with other miRs, mRNA, DNA, or proteins, miRs can modulate responses to chemotherapy and radiotherapy. Although miRs have great potential as predictive biomarkers in guiding precision medicine or as therapeutic targets to improve chemosensitivity and radiosensitivity in the treatment of CRC and mCRC, the use of miRs in clinical practice warrants further investigation.

## Figures and Tables

**Figure 1 cancers-15-01358-f001:**
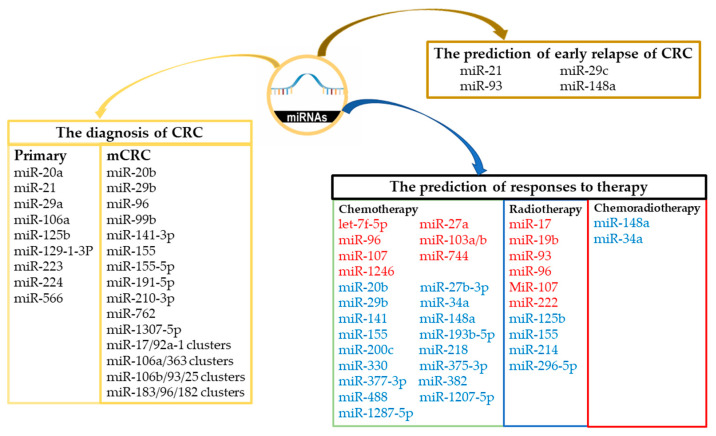
MiRs associated with the detection of CRC and mCRC, the prediction of early relapse of CRC, and the prediction of responses to chemotherapy/radiotherapy/chemoradiotherapy in CRC and mCRC. Red words indicate the oncomiRs enhance chemoresistance/radioresistance and blue words indicate the anti-oncomiRs that enhance chemosensitivity/radiosensitivity in CRC and mCRC.

**Table 1 cancers-15-01358-t001:** MiRs used in the detection of CRC and mCRC.

MiRs Used in the Diagnosis of Primary CRC		Sample Source	Ref.
miR-20a	MiR-20a was upregulated in patients with CRC (relative to controls) and may be a valid biomarker for CRC detection but may not be a strong prognostic indicator.	Feces, serum, and tumor tissue	[62]
miR-21	The high expression of miR-21 was significantly correlated with advanced clinical stage and poor cell differentiation.	Tumor tissue	[64]
miR-29a, miR-223, miR-224	The expression levels of miR-29a, miR-223, and miR-224 from patients with CRC were significantly lower than those from health volunteers.	Feces	[61]
miR-106a, miR-125b	MiR-106a and miR-125b were associated with the pathogenesis and invasion of CRC and may be used as significant prognostic markers of early-stage CRC.	Tumor	[63]
miR-129-1-3p, miR-566	A urinary biomarker panel combining miR-129-1-3p and miR-566 could accurately detect stage 0/I CRC.	Urinary samples	[48]
**MiRs Used in the Prediction of Early Relapse of CRC**		**Sample Source**	**Ref.**
miR-21	Lower serum miR-21 expression was associated with higher local recurrence (*p* = 0.025) and mortality (*p* = 0.029).	Serum	[65]
miR-29c	miR-29c expression in the early relapse group was significantly lower than that in the non–early relapse group.	Tumor tissue	[66]
miR-93	The miR-93 expression levels of the early relapse group were significantly lower than those of the non–early relapse group. The in vitro and in vivo effects of miR-93 overexpression were inhibited by CRC proliferation and migration, and miR-93 decreased CRC recurrence.	Tumor tissue	[60]
miR-148a	miR-148a expression levels in the early relapse group were significantly lower than those in the non–early relapse group. MiR-148a inhibits VEGF secretion by indirectly targeting hypoxia-inducible factor 1 subunit alpha (HIF-1α).	Tumor tissue and serum	[67,68]
	Lower miR-148a expression was positively associated with advanced TNM stage, poor tumor differentiation, lymph node metastasis, and distant metastasis.	Tumor tissue	[69]
**MiRs Used in the Diagnosis of mCRC**		**Sample Source**	**Ref.**
miR-17/92a-1, miR-106a/363, miR-106b/93/25, miR-183/96/182 clusters	The miR-17/92a-1, miR-106a/363, miR-106b/93/25, and miR-183/96/182 clusters were strongly associated with metastasis and poor patient survival.	Tumor tissue, blood, and feces	[70]
miR-20b, miR-29b, miR-155	A multivariate analysis of patients with mCRC receiving bevacizumab-based treatment revealed that circulating expression levels of miR-20b, miR-29b, and miR-155 were significantly associated with progression-free survival (*p* < 0.05) and overall survival (*p* < 0.05).	Serum	[71]
miR-96/miR-99b	Plasma miR-96/miR-99b expression levels may serve as a promising biomarker for the early detection of mCRC.	Plasma	[72]
miR-210-3p, miR-191-5p, miR-141-3p, miR-1307-5p, miR-155-5p	Five miRNAs—miR-210-3p, miR-191-5p, mir-141-3p, miR-1307-5p, and miR-155-5p—were determined to be upregulated at multiple metastatic sites according to an analysis of new and previously published next-generation sequencing data sets of samples of primary CRC and mCRC (liver, lung, and peritoneal metastases) and tumor-adjacent tissues.	Tumor tissue	[73]
miR-762	The circulating miR-762 levels of patients with CRC with distant metastasis were higher than those of patients with CRC without distant metastasis.	Serum	[74]

**Table 2 cancers-15-01358-t002:** OncomiRs enhance chemoresistance in patients with CRC or mCRC.

Family	miRNAs	Verified Targets in CRC or Other Cancers	Sample	Target for miRNA	Ref.
Let-7	Let-7a/b/c/d /e/f/g/i, miR-98	Upregulation of let-7f-5p promotes chemoresistance in CRC by increasing the expression levels of the antiapoptotic proteins B-cell lymphoma 2 (Bcl-2) and B-cell lymphoma extra-large (Bcl-xL) and decreasing the activity of caspase-3 and caspase-9 in CRC cells.	Tumor tissue	p53, p53-inducible nuclear protein 1, p53-inducible nuclear protein 2 and caspase-3.	[84,85]
miR-27	miR-27a, miR-27b	MiR-27a-overexpressing hampered *AMPK*, enhanced mTOR signaling, unrestricted cell growth, and enhanced chemoresistance.	Tumor tissue	*AMPK*	[88]
miR-96	miR-96	Inhibition of miR-96 enhances the sensitivity of CRC cells to oxaliplatin.	Serum	*TPM1*	[86]
miR-103	miR-103a/b, miR-107	MiR-107 induces chemoresistance in CRC cells.	Tumor tissue	CAB39-AMPK-mTOR pathway	[89]
		MiR-103 and miR-107 enhance chemoresistance in CRC cells by promoting cell stemness.		Wnt/β-catenin signaling	[90]
miR-744	miR-744	The expression levels of miR-744 were significantly elevated in CRC tissues from patients who exhibited preoperative oxaliplatin chemoresistance. MiR-744 may positively mediate oxaliplatin chemoresistance.	Tumor tissue	*BIN1*	[92]
miR-1246	miR-1246	MiR-1246 was overexpressed in CD44v6+ cells and associated with poor overall survival and disease-free survival in patients with CRC. CD44v6+ cells exhibited elevated resistance to chemotherapeutic drugs and significantly higher tumor initiation capacity.	Tumor tissue	*DENN/MADD Domain Containing 2D* (*DENND2D*)	[93]

**Table 3 cancers-15-01358-t003:** Anti-oncomiRs enhance chemosensitivity in patients with CRC or mCRC.

Family	miRNAs	Verified Targets in CRC or Other Cancers	Sample Source	Target for miRNA	Ref.
miR-8	miR-8, miR-141, miR-200a/b/c, miR-429	MiR-141-3p enhanced the cetuximab sensitivity of CRC cells	Tumor tissue	*ZEB1-ZEB2*	[94,95]
		Expression of miR-200c and miR-141 was downregulated in oxaliplatin-resistant CRC cell lines.		*EGFR*	[96]
miR-27	miR-27a, miR-27b-3p	MiR-27b-3p sensitizes CRC cells to oxaliplatin in vitro and in vivo, and miR-27b-3p expression was positively correlated with disease-free survival time in patients with CRC.	Tumor tissue	*ATG10*	[102]
miR-29	miR-29a, miR-29b, miR-29c	Circulating miR-20b, miR-29b, and miR-155 expression levels were significantly associated with progression-free survival (*p* < 0.05) and overall survival (*p* < 0.05).	Serum	No data	[71]
miR-34	miR-34a/b/c/d	MiR-34a enhanced chemosensitivity to 5-FU.	Serum	*E2F3*; *SIRT1*.	[87]
miR148	miR-148a/b, miR-152	MiR-148a suppressed the expression of stem cell markers and increased chemosensitivity, cell invasion, and cell migration.	Tumor tissue	WNT10b and beta-catenin signaling pathway.	[69]
		MiR-148a decreased angiogenesis and increased CRC cell apoptosis by downregulating HIF-1α/VEGF and Mcl-1, and serum miR-148a levels have prognostic or predictive value in patients with mCRC receiving bevacizumab.			[103]
miR-154	miR-154, miR-323a, miR-369-3p, miR-377, miR-381, miR-382, miR-409, miR-410	MiR-377-3p expression levels in CRC samples (especially those from patients with stage III/IV CRC) were significantly lower than those in normal mucosa tissues. Overexpression of miR-377-3p enhanced the chemosensitivity of CRC cells by inhibiting Wnt/beta-catenin signaling by directly targeting *ZEB2* and *XIAP*, which are positive regulators of Wnt/β-catenin signaling.	Tumor tissue	*ZEB2* and *XIAP*	[97]
		MiR-382 functions as a tumor suppressor and chemosensitizer in CRC.			[100]
miR-155	miR-155	Circulating expression levels of miR-20b, miR-29b, and miR-155 were significantly associated with progression-free survival (*p* < 0.05) and overall survival (*p* < 0.05).	Serum	No data	[71]
		MiR-155 induced radioresistance by targeting *FOXO3a*.		*FOXO3a*.	[104]
miR-193	miR-193a/b	MiR-193b-5p enhanced chemosensitivity to 5-FU.	Tumor tissue	HMGA2/MAPK pathway	[98]
		CRC tissues and adjacent noncancerous tissues were obtained from 67 patients who had undergone surgery. Upregulation of miR-193-5p, particularly in combination with 5-FU and oxaliplatin, reduced the expression levels of *CXCR4*. A miR-193a-5p *mimic* suppressed *CXCR4*-induced CRC cell proliferation.		*CXCR4*.	[99]
miR-218	miR-218-1/2	MiR-218 enhanced 5-FU cytotoxicity by suppressing thymidylate synthase and MiR-218 promoted apoptosis, inhibited cell proliferation, and caused cell cycle arrest	Tumor tissue	thymidylate synthase; *BIRC5*	[105]
miR-330	miR-330	MiR-330 inhibited CRC cell proliferation and enhanced CRC cell chemosensitivity to 5-FU	Tumor tissue	*Hexokinase 2* *Thymidylate synthase*	[30] [106]
miR-375	miR-375-3p	MiR-375 enhanced CRC cell chemosensitivity to 5-FU by directly targeting *YAP1* and *SP1*.	Tumor tissue	*YAP1* and *SP1*	[107]
		MiR-375 enhanced CRC cell chemosensitivity to 5-FU by targeting thymidylate synthase.	Tumor tissue	thymidylate synthase	[108]
miR-488	miR-488	MiR-488 mimics transfected into CRC cell lines induced decreases in glucose uptake and increases in oxaliplatin/5-FU chemosensistivity.	Serum	*PFKFB3*	[109]
miR-1207	miR-1207-5p	Upregulation of miR-1207-5p inhibited bevacizumab resistance in CRC cells.	Tumor tissue	*ABCC1*.	[110]
miR-1287	miR-1287-5p	Lower miR-1287-5p expression levels upregulate the mRNA expression of *Y-box binding protein 1* (*YBX1*) and protein levels of YBX1, thereby inducing CRC cell proliferation and migration.	Tumor tissue	*YBX1*	[111]
		The multifunctional *YBX1* is overexpressed and phosphorylated in CRC and is associated with cetuximab resistance.			[112]
miR-1915	miR-1915	Exosomal delivery of miR-1915-3p can improve the chemosensitivity of oxaliplatin by suppressing the epithelial–mesenchymal transition.	Plasma	*PFKFB3* and *USP2*	[101]

**Table 5 cancers-15-01358-t005:** Anti-oncomiRs that enhance radiosensitivity in CRC and mCRC.

Family	MiRs	Verified Targets in CRC or Other Cancers	Sample Source	Target for miRNA	Ref.
miR-10	miR-10a/b, miR-99a/b, miR-100, miR-125a/b-1/b-2	MiR-125b was highly expressed both in tissues and serum obtained from nonresponders to CRT. Circulating miR-125b levels exhibited greater discriminatory power for treatment responses than did serum CEA levels.	Serum	No data	[137]
miR-155	miR-155	Circulating miR-20b, miR-29b, and miR-155 levels were significantly associated with progression-free survival (*p* < 0.05) and overall survival (*p* < 0.05). MiR-155 induced radiation resistance.	Serum	No data *FOXO3a*	[71] [104]
miR-214	miR-214	MiR-214 promoted radiosensitivity by inhibiting *ATG12*-mediated autophagy in CRC.	Tumor tissue	*ATG12*	[139]
miR-296	miR-296-5p	MiR-296-5p enhanced the radiosensitivity.	Serum	*IGF1R*; *MSI1*	[141] [142]

**Table 6 cancers-15-01358-t006:** MiRs involved in chemoradiosensitivity or chemoradioresistance in CRC and mCRC.

Anti-OncomiRs That Enhance Chemoradiosensitivity					
Family	MiRs	Verified Targets in CRC or Other Cancers	Sample Source	Target for miRNA	Ref.
miR-148	miR-148a/b, miR-152	MiR-148a enhances the chemoradiosensitivity of patients with rectal cancer.	Tumor tissue	*c-Met*	[113]
miR-34	miR-34a/b/c/d	MiR-34a attenuates the chemoresistance of colon cancer to 5-FU by inhibiting *E2F3* and *SIRT1*. The miR-34a mimic *MRX34* is the first synthetic miRNA to undergo clinical trials.	Serum	*E2F3*; *SIRT1*	[87,143]

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
