# Peer review of "MicroRNAs as Predictive Biomarkers in Patients with Colorectal Cancer Receiving Chemotherapy or Chemoradiotherapy: A Narrative Literature Review"

_cancers, 2023, doi:10.3390/cancers15051358_

Round 1
Reviewer 1 Report (Previous Reviewer 1)
WELL DONE
Author Response
Thank you for your affirmation.
Reviewer 2 Report (New Reviewer)
The authors have presented their manuscript titled: MicroRNAs as Predictive Biomarkers in Patients with Colorectal Cancer Receiving Chemotherapy or Chemoradiotherapy: A Narrative Literature Review
This review provides a comprehensive assessment of miRs signatures in CRC and mCRC. I find the manuscript is well written except for a few complex sentences. The authors have covered and cited appropriate literature. However, I would like to raise a few comments for the authors.
Authors have section subheadings like MiRs Used in the Detection of CRC and mCRC and MiRs Used in the Prediction of Responses to Radiotherapy in CRC. I think these should be stated as miRs associated with or have the potential to be used for detection since authors have not cited any clinical assay/treatment that uses miRs for detection/prediction.
Following the previous comment, authors have suggested potential roles for these miRs in clinical settings. If possible, I think it would be great to have a section that covers current undergoing clinical trials for miRs. For example, authors have mentioned one such instance: MRX34, a miR-34a mimic, is the first synthetic miR to undergo clinical trials to restore the sensitivity of CRC cells to chemotherapeutic agents.
It would be helpful to have a summary fig that can abstractly show different mechanisms for these miRNAs and their utility in CRC diagnosis and assessment.
The first sentence, in Conclusion, implies authors have conducted biomarker prediction in the paper, which is not the case. Please rephrase this along the lines of that other studies have found miRs to be predictive biomarkers. Also, this sentence is complex; please break it into smaller, simpler sentences.
Author Response
Please see the attachment.

Reviewer 3 Report (New Reviewer)
It would be better to include figures, in which improve the scientific value of the paper and optimize the level of observation. Moreover, figure provides a better way to interpret data gathered in this review for readers.
Author Response
Please see the attachment.

This manuscript is a resubmission of an earlier submission. The following is a list of the peer review reports and author responses from that submission.
Round 1
Reviewer 1 Report
EXCELLENT WORK.FULL OF IMPORTANT INFORMATION AND AT THE SAME TIME LACONIC!THE ONLY THING THAT MISSES IS A REFER TO THE CLASSIFICATION OF THE CRC AND THE POSSIBLE CORRELATION OF THE CA TYPES WITH THE miRNA.
Reviewer 2 Report
1. The current review on the role of microRNAs in colorectal cancer is interesting. However similar reviews have already been published in the past including https://pubmed.ncbi.nlm.nih.gov/29156521/, https://pubmed.ncbi.nlm.nih.gov/28902152/, https://pubmed.ncbi.nlm.nih.gov/25206276/, https://doi.org/10.1016/B978-0-12-819937-4.00008-X and several others. As such the novelty and significance of this review to the field of CRC is missing. Authors must emphasize the relevance of this review inn comparison to the previous published review.
2. Authors should add a section emphasizing on the application of these CRC specific miRNAs in clinics.
Reviewer 3 Report
Yang and colleagues developed a comprehensive review on biomarkers that can either help on the diagnose, or to predict the therapeutic resistance, of CRC patients. This is very timely and relevant subject, and this manuscript could be a useful guide for future research. Nevertheless, the article needs to be improved.
A connexion, between the tables and the text, must exist. The reader is overwhelmed (and even confused) with so much written information.
In addition, some references at the table are not at the text, and vice-versa.
To avoid having such long tables, perhaps some tables could be splitted.
Perhaps sub-titles could help in 3.1; however, it is not clear 3.3 is all about mCRC.
The sample type (where the miRNA was detected) should be added as a column to the tables, at least when it may vary in the same table.
If a target for the miRNA (or panel of miRNA) was identified, a column with that information could be useful.
In some cases the text in the tables should be simplified.
The conclusion should be based on the manuscript "results".
Minor:
Material and Methods do not exist for a review of the literature.
All gene symbols and abbreviations must be shown in full the first time they appear in the text (ex.: RAS, BRAF, lncRNA, etc).
Indicate country for the HPA (line 59).
Using the terminology of "downstream proteins" is inaccurate for the resultant molecule of the mRNA translation (line 107).
Word missing in line 48, between "involving" and "revealed".
References should be added in line 190 (after signals), line 224 (after group), line 278 (after pathways).
Justify the use of ref 114 in line 279.
Reviewer 4 Report
The submitted paper is a review article. However, 10 authors is not a proper paper.
It cannot be determined that 10 authors are required to complete this review article.
Including many authors to get credit for a paper is totally inappropriate for a scientific theory.
The submitted review article is a list of selected papers and are not attractive.
Contribution to the field of colorectal cancer/microRNA research is extremely limited.
Round 2
Reviewer 2 Report
1. The relevance of this review is still lacking. Please emphasize clearly the relevance of this review in comparison to the previous published reviews.
2. Extensive english editing is required.
Author Response
The relevance of this review is still lacking. Please emphasize clearly the relevance of this review in comparison to the previous published reviews.
Reply: Thank for the reviewer’s advice. We added the sentences as “Previous several reviews about miRs as biomarkers in CRC have already been published [53-55] and even written into text book to discuss the potential cancer sensitizing agents for chemotherapy of CRC [56]. Stiegelbauer et al., summarizes the chemotherapeutic approaches for treating CRC and highlights the potential role of miRNAs as novel predictive biomarkers for chemo-resistant in CRC patients [53]. Masuda et al., through a meta-analysis proved that miRs having strong statistical confidence as biomarkers in CRC but they also point out that most miRs reports were small-scale studies [54]. Shirafkan et al., summarizes the roles of miRs in CRC by emphasizing their importance in different signaling pathways such as EGFR pathway, transforming growth factor beta (TGF-ß) and the tumor protein (TP53) network and suggested miRs as predictive factors of chemotherapy [55]. In this review, we also discuss some miR-relative pathways and then we focus on these miRs can serve as clinical biomarkers in screening or detection of CRC or mCRC or CRC recurrence prevention. We summarize these miRs can drivers and modulators of CRC resistance of chemotherapy or chemoradiotherapy and may be potential genomic therapies in the future.” (line 126-140), of which is something different from previous published review.
- Stiegelbauer, V.; Perakis, S.; Deutsch, A.; Ling, H.; Gerger, A.; Pichler, M. MicroRNAs as novel predictive biomarkers and therapeutic targets in colorectal cancer. World J Gastroenterol 2014, 7, 11727-11735, doi:10.3748/wjg.v20.i33.11727.
- Masuda, T.; Hayashi, N.; Kuroda, Y.; Ito, S.; Eguchi, H.; Mimor, K. MicroRNAs as Biomarkers in Colorectal Cancer. Cancers 2017, 9, doi:10.3390/cancers9090124.
- Shirafkan, N.; Mansoori, B.; Mohammadi, A.; Shomali, N.; Ghasbi, M.; Baradaran, B. MicroRNAs as novel biomarkers for colorectal cancer: New outlooks. Biomed Pharmacother 2018, 97, 1319-1330, doi:10.1016/j.biopha.2017.11.046.
- WaiHon, K.; Othman, N.; Hanif, E.A.M.; Nasir, S.N.; AbdRazak, N.S.; Jamal, R.; Abu, N. Cancer Sensitizing Agents for Chemotherapy; 2020; Volume 8 pp. 135-151.
Extensive English editing is required.
Reply: Thank for the reviewer’s advice. English-editing with the certificate is enclosed in the attached file.

Reviewer 3 Report
The authors have updated their review. Unfortunately, this reviewer feels the previous comments were not all fullfilled.
In addition, it seems the authors eliminated some miRNAs from the tables (miR-1287, miR-21, miR-29), as well as moved from one table to another (miR-29c), without giving notice of change to the reviewers and/or explaining the reason.
Author Response
1. The authors have updated their review. Unfortunately, this reviewer feels the previous comments were not all fullfilled.
Reply: Thank for the reviewer’s advice. We found that some full gene names was not amended and added into the text follows:
AMP-activated protein kinase- mammalian target of rapamycin (AMPK-–mTOR) (line 229)
forkhead box M1- ATP binding cassette subfamily C member 5 (FOXM1-ABCC5/10) (line 247)
6-phosphofructo-2-kinase/fructose-2,6-biphosphatase 3 (PFKFB3) (line 250)
ubiquitin specific peptidase 2(USP2) (line 251)
Sp1 transcription factor (SP1) (line259)
MET proto-oncogene, receptor tyrosine kinase (c-Met) (line 266)
MYC proto-oncogene, bHLH transcription factor (c-Myc) (line 270)
6-phosphofructo-2-kinase/fructose-2,6-biphosphatase 3(PFKFB3) (line276)
Y-box binding protein 1 (YBX1) (line 299)
hypoxia inducible factor 1 subunit alpha (HIF-1α) (line 305)
MCL1 apoptosis regulator, BCL2 family member (Mcl-1) (line306)
phosphatase and tensin homolog/phosphatidylinositol-3 kinase / AKT serine/threonine kinase 1 (PTEN/PI3K/AKT) (line327)
forkhead box A1 (FOXA1) (line330)
transforming growth factor beta 3 (TGFB3) (line331)
glypican 3 (GPC3)(line334)
F-box and WD repeat domain containing 7(FBXW7) (line335)
autophagy related 12 (ATG12) (line349)
insulin like growth factor 1 receptor (IGF1R) (line354)
musashi RNA binding protein 1 (MSI1) (line355)
2. In addition, it seems the authors eliminated some miRNAs from the tables (miR-1287, miR-21, miR-29), as well as moved from one table to another (miR-29c), without giving notice of change to the reviewers and/or explaining the reason.
Reply: Thank for the reviewer’s remind and the miR-1287 is put back to the table 3. miR-21 and miR-29 were changed the location in Table due to (1) The miR-21 expression levels of CRC tumor tissue and serum shown different clinical diagnosis meanings. We replaced miR-21 into the tables 1; (2). The miR-29c expression levels from single paper was mentioned at two parts: “MiRs used in the prediction of early relapse of CRC” and “MiRs used in the diagnosis of mCRC” in the table 1. Consider to simplify the tables 1, we replace it in “MiRs used in the prediction of early relapse of CRC” part.

Reviewer 4 Report
Gift authorship should be taken very seriously.
Author Response
Gift authorship should be taken very seriously.
The authorship is confirmed as follows: Author Contributions: Conceptualization, J.-Y. Wang; methodology, I-P.Yang., K.-L.Yip and Y.-T.Chang; formal analysis, I-P.Yang., K.-L.Yip and Y.-T.Chang; writing - original draft preparation, I-P.Yang; writing-review and editing, Y.-C. Chen, C.-W. Huang, H.-L. Tsai, Y.-S. Yeh, and J.-Y. Wang; visualization, I-P.Yang., K.-L.Yip and Y.-T.Chang; supervision, J.-Y. Wang; funding acquisition, J.-Y. Wang. All the authors have read and agreed to the published version of the manuscript. (line 382-386)

Round 3
Reviewer 2 Report
Unfortunately the current study does not add much to the field as similar reviews have already being published. The significance of this review as compared to the previous ones is still not clear. Moreover the English language is not appropriate for publication.